# Larger Vertical Ridge Augmentation: A Retrospective Multicenter Comparative Analysis of Seven Surgical Techniques

**DOI:** 10.3390/jcm14124284

**Published:** 2025-06-16

**Authors:** Andreas Pabst, Abdulmonem Alshihri, Philipp Becker, Amely Hartmann, Diana Heimes, Eleni Kapogianni, Frank Kloss, Keyvan Sagheb, Markus Troeltzsch, Jochen Tunkel, Christian Walter, Peer W. Kämmerer

**Affiliations:** 1Department of Oral and Maxillofacial Surgery, German Armed Forces Central Hospital, Rübenacherstr. 170, 56072 Koblenz, Germany; andipabst@me.com (A.P.); becker-ph@web.de (P.B.); 2Department of Oral and Maxillofacial Surgery, University Medical Center Mainz, Augustusplatz 2, 55131 Mainz, Germany; amelyhartmann@web.de (A.H.); diana.heimes@unimedizin-mainz.de (D.H.); keyvan.sagheb@unimedizin-mainz.de (K.S.); christian.walter@unimedizin-mainz.de (C.W.); 3Department of Prosthetic Dental Sciences, College of Dentistry, King Saud University, P.O. Box 60169, Riyadh 11545, Saudi Arabia; monem.alshihri@post.harvard.edu; 4Private Practice for Oral Surgery and Periodontology, Volmarstr. 8, 70794 Filderstadt-Bernhausen, Germany; 5Private Practice for Oral Surgery and Periodontology, Fasanenstraße 81, 10623 Berlin, Germany; welcome@city-chirurgie.de; 6Private Practice for Oral and Maxillofacial Surgery, Kärtnerstraße 62, 9900 Lienz, Austria; info@mkg-kloss.at; 7Private Practice for Oral and Maxillofacial Surgery, Maximilianstr. 5, 91522 Ansbach, Germany; troeltzsch@gmx.net; 8Private Practice for Oral Surgery and Periodontology, Königstraße 19, 32545 Bad Oeynhausen, Germany; j.tunkel@fachzahnarzt-praxis.de; 9Private Practice for Oral and Maxillofacial Surgery, Haifa-Allee 20, 55128 Mainz, Germany

**Keywords:** vertical ridge augmentation, alveolar ridge augmentation, autogenous bone graft, allogeneic bone graft, CAD/CAM titanium mesh, magnesium scaffold, donor site morbidity, bone regeneration, graft resorption, bone gain

## Abstract

**Background:** Vertical alveolar ridge augmentation (ARA) > 3 mm is associated with increased surgical complexity and higher complication rates. Despite the availability of various ARA techniques and graft materials, robust comparative clinical data remain limited. This retrospective multicenter study aimed to evaluate and compare surgical and patient-relevant outcomes across seven established vertical ARA techniques. **Methods:** This retrospective multicenter study included 70 cases of vertical ARA > 3 mm using seven different techniques (10 cases each): an iliac crest graft (ICG), intraoral autogenous bone block (IBB), allogeneic bone block (ABB), CAD/CAM ABB, CAD/CAM titanium mesh (CAD/CAM TM), magnesium scaffold (MS), and the allogeneic shell technique (ST). The outcome parameters included harvesting and insertion time, bone gain (vertical and horizontal, after a minimum of one year), graft resorption (after one year), donor site morbidity, dehiscence rate, need for material removal, and biological and general financial costs. **Results:** Harvesting time significantly varied among the different ARA techniques (*p* = 0.0025), with the longest mean durations in ICGs (51.6 ± 5.8 min) and IBBs (36.5 ± 10.8 min), and no harvesting was required for the other techniques. Insertion times also significantly differed between the different ARA techniques (*p* < 0.0001) and were longest in IBBs (50.1 ± 7.5 min) and the ST (47.3 ± 13.9 min). ICGs achieved the highest vertical and horizontal bone gain (5.6 ± 0.4 mm), while ABBs and CAD/CAM ABBs showed the lowest (~3.0 mm). Resorption rates significantly differed between the different ARA techniques (*p* < 0.0001) and were highest for ICGs (25.9 ± 3.9%) and lowest for MSs (5.1 ± 1.5%). Donor site morbidity was 100% in ICGs and 50% in IBBs, with no morbidity in the other groups. Dehiscence rates were 10% in most techniques but 30% in CAD/CAM TMs. Removals were required in all techniques except MSs. Biological and financial costs were high for ICGs and CAD/CAM ABBs and low for MSs. **Conclusions:** Vertical ARA techniques significantly differ regarding harvesting and insertion time, bone gain, graft resorption, donor site morbidity, dehiscence rates, removals, and costs. While ICGs achieved the highest bone volume, less invasive techniques, such as CAD/CAM-based or resorbable scaffolds, reduced biological costs and complication risks. Technique selection should be individualized based on defects, patients, and reconstructive goals.

## 1. Introduction

Alveolar ridge augmentation (ARA) is used to reconstruct the atrophic alveolar ridge following tooth loss, as sufficient vertical and horizontal bone dimensions are critical for dental implant placement. Comparable implant survival rates have been reported for augmented and non-augmented sites, even after more than 10 years of follow-up [1]. Various ARA techniques, graft materials, and bone substitutes (BSs) are available, each with distinct surgical demands, biological properties, and limitations. Autogenous bone grafts, such as iliac crest grafts (ICGs) and intraoral bone blocks (IBBs), are still considered the gold standard for ARA [2,3]. In contrast, allogeneic grafts, including allogeneic bone blocks (ABBs) and cortical plates used in the shell technique (ST), have shown comparable clinical outcomes [4,5]. Various BSs, such as xenografts, allografts, and synthetic substitutes, can provide multiple reconstructive options. Their regenerative potential can be further enhanced by biologicals such as platelet-derived growth factor (PDGF) and autogenous platelet concentrates (APCs), particularly platelet-rich fibrin (PRF) [2,6,7]. An innovative approach combines BSs with biologicals, additives, and APC, such as collagen, hyaluronic acid (HA), or PRF, to further enhance their regenerative potential [8,9,10,11]. Nevertheless, preventive strategies in ARA, such as ridge preservation or short implants in the posterior maxilla to avoid sinus floor elevation, are becoming increasingly common [12,13,14,15].

Personalized ARA concepts have gained increasing popularity in recent years [2]. Advances in computer-aided design (CAD) and computer-aided manufacturing (CAM) have enabled the development of customized grafts and scaffolds, such as CAD/CAM ABB and titanium meshes (TMs) [16,17,18,19]. These ARA techniques provide a precise fit and stable structural support for predictable bone regeneration. Unlike other ARA techniques, such as guided bone regeneration (GBR) with collagen membranes, wound closure in CAD/CAM ABB and TM procedures does not compromise the augmentative volume [20]. However, clinical data on their long-term success remains limited. A key limitation of CAD/CAM TMs is the need for mesh removal before implant placement. As an alternative, biodegradable magnesium scaffolds (MSs) and fixation screws have been developed to provide stable structural support without the need for scaffold and screw removal [21,22]. However, hydrogen gas release has been observed during MS degradation, potentially leading to transient wound expansion [23]. A metallic taste has also been reported in isolated cases [23]. Despite these observations, current clinical data are insufficient to allow for a comprehensive evaluation of MS performance.

The amount of vertical and horizontal bone gain achieved through different ARA techniques and grafts is a key determinant of clinical success. A systematic review by Troeltzsch et al. reported an average horizontal bone gain of 3.7 ± 1.2 mm and vertical bone gain of 3.7 ± 1.4 mm when using particulate grafts and BSs [24]. The review also found that block grafts enhance horizontal bone gain, while stable structural supports, such as TMs, are particularly effective in achieving vertical bone augmentation [24]. However, excessive contouring beyond the skeletal envelope, for example, with ABBs, may increase the risk of complications such as graft exposure or soft tissue dehiscence [5]. Beyond bone gain alone, choosing an appropriate ARA technique and graft requires consideration of multiple factors, including donor site morbidity, surgical duration (harvesting and insertion time), dehiscence rates, graft resorption, material removal, and biological and general financial costs.

This retrospective multicenter study evaluated patients who underwent vertical ARA > 3 mm using seven different ARA techniques and graft types: ICG, IBB, ABB, CAD/CAM ABB, TM, MS, and the ST. The ARA techniques were compared regarding surgical time (harvesting and insertion), bone gain (vertical and horizontal), graft resorption, dehiscence rates, material removal, and biological and general financial costs. The aim was to assess the clinical potential and limitations of each approach.

## 2. Materials and Methods

### 2.1. Study Design and Patient Selection

Seventy cases of ARA were retrospectively analyzed from eight surgical centers, including two departments of Oral and Maxillofacial Surgery (OMFS), three OMFS private practices, and three private practices for Oral Surgery. This study was conducted in accordance with the Declaration of Helsinki and was approved by the State Medical Association Ethics Committee of Rhineland-Palatinate (Mainz, Germany; approval no. 2023-17326).

Each of the seven ARA techniques and grafting approaches was represented by 10 cases: iliac crest graft (ICG), autogenous intraoral bone block (IBB), allogeneic bone block (ABB), customized CAD/CAM allogeneic bone block (CAD/CAM ABB), customized CAD/CAM titanium mesh (CAD/CAM TM), magnesium scaffold (MS), and the allogeneic shell technique (ST). All cases involved vertical ARA > 3 mm. Surgical procedures were performed by experienced full-board-certified OMF surgeons or oral surgeons. These seven ARA techniques were selected because they encompass commonly used clinical methods that represent a wide range of surgical approaches, invasiveness, material characteristics, and clinical relevance in contemporary OMFS. The 10 cases per technique were selected based on the availability of complete documentation and clinical comparability within the retrospective timeframe from 2017 to 2022. This timeframe ensured a minimum one-year follow-up after implant placement, reflecting recent clinical practice standards across the participating centers.

The inclusion criteria comprised patients with alveolar ridge atrophy who underwent ARA (>3 mm in the vertical dimension) before implant placement, with complete clinical documentation including preoperative assessment, radiographs, surgical reports, postoperative records, and follow-up data (including a CBCT) for at least one year after implant placement. Cases with incomplete documentation were excluded. To minimize variability and enhance comparability, this retrospective analysis included only cases with comparable patient characteristics, defect types, and augmentation requirements, as well as complete documentation and a minimum of one year of follow-up. Exclusion criteria included cases with incomplete documentation, cases with less than one year of follow-up, and patients with severe systemic conditions that contraindicated oral surgery or bone augmentation (such as uncontrolled diabetes, severe immunosuppression, or bisphosphonate-related osteonecrosis of the jaw). Table 1 summarizes the demographic and clinical parameters (age, sex, BMI, smoking status, anatomical location, and implant placement approach) for each ARA technique group, providing an overview of the patient cohort.

The following parameters were assessed: surgical time (harvesting and insertion), bone gain (vertical and horizontal), graft resorption, donor site morbidity, soft tissue dehiscence, need for material removal (e.g., screws or meshes), and biological and general financial costs.

### 2.2. Surgical Techniques

Figure 1, Figure 2, Figure 3, Figure 4, Figure 5, Figure 6 and Figure 7 illustrate clinical cases for each ARA technique, focusing on the general surgical procedure for each technique rather than the consistent presentation of vertical ridge augmentation cases. The following approaches were included:ICG—Autogenous bone block harvested from the iliac crest and fixed with titanium screws (Figure 1).IBB—Autogenous bone block harvested intraorally from the mandibular angle and fixed with titanium screws (Figure 2).ABB—Prefabricated, processed allogeneic cancellous bone block (maxgraft^®^ blocks; botiss biomaterials GmbH, Zossen, Germany), intraoperatively shaped to fit the defect and fixed with titanium screws (Figure 3).CAD/CAM ABB—Customized, patient-specific CAD/CAM-manufactured allogeneic cancellous bone block (maxgraft^®^ bonebuilder; botiss), designed from Cone-beam computed tomography (CBCT) data and fixed with titanium screws (Figure 4).CAD/CAM TM—Customized CAD/CAM-manufactured titanium mesh (Yxoss CBR^®^; ReOss GmbH, Filderstadt, Germany), designed from CBCT data and filled with autogenous bone chips, allogeneic granules (maxgraft^®^ granules; botiss), and/or bovine BS (±hyaluronic acid), and fixed with titanium screws (Figure 5).MS—Biodegradable magnesium scaffold (NOVAMag^®^; botiss), filled with autogenous bone chips, allogeneic granules, and/or bovine BS (±hyaluronic acid), and fixed with biodegradable magnesium screws (Figure 6).ST—Prefabricated, processed allogeneic cortical bone plates (maxgraft^®^ cortico; botiss), intraoperatively customized to fit the defect, filled with autogenous bone chips, allogeneic granules, and/or bovine BS (±hyaluronic acid), and fixed with titanium screws (Figure 7).

In some cases, the augmented areas were covered with porcine collagen and/or PRF membranes. Wound closure was performed tension-free using monofilament, non-resorbable suture material. All surgical procedures followed standardized aseptic protocols and were performed under local anesthesia, sedation, or general anesthesia (e.g., for ICG harvesting). Standard preoperative prophylaxis included antibiotics (e.g., amoxicillin 2 g or clindamycin 600 mg orally) and preemptive analgesia (e.g., ibuprofen 600 mg orally).

### 2.3. Postoperative Care and Follow-Up

All patients received standardized postoperative care, including analgesics (e.g., ibuprofen 400–600 mg and paracetamol 500 mg, orally), antibiotics (e.g., amoxicillin 2 g or clindamycin 600 mg, orally), and chlorhexidine mouth rinses. Sutures were removed after 7 to 10 days. Clinical follow-up appointments were scheduled at 1, 3, and 6 months postoperatively. CBCT scans were obtained at baseline and before implant placement to evaluate bone gain and support implant planning. For inclusion in this study, an additional CBCT at least one year after implant placement was needed.

### 2.4. Data Collection and Outcome Measures

The following parameters were assessed for each ARA technique and graft type: Harvesting and insertion times were recorded separately for the respective surgical steps. CBCT scans and intraoperative evaluation during surgical re-entry and one year after implant placement assessed bone gain in vertical and horizontal dimensions. Graft resorption was measured based on volumetric changes observed in CBCT imaging and intraoperative findings, one year after implant insertion, radiographically. Donor site morbidity was evaluated based on postoperative pain, swelling, infection, wound dehiscence, or sensory disturbances. The need for removals, such as explantation of grafts, scaffolds, or fixation screws, was documented. The biological cost was estimated, including surgical invasiveness, donor site morbidity, complication risk, and healing duration. The general financial cost was approximated based on surgery-related and material-related expenses. We emphasize that these estimations are descriptive and qualitative and were not quantified using standardized or validated assessment scales due to the retrospective study design.

### 2.5. Statistics

All data were analyzed using descriptive and inferential statistical methods. Descriptive statistics included the calculation of means, standard deviations, and ranges for quantitative variables, and frequencies for categorical data. For comparisons of quantitative variables between the different ARA techniques, we applied Kruskal–Wallis tests (non-parametric) due to the distribution of the data. Where significant overall differences were found, post-hoc pairwise Mann–Whitney U tests with Bonferroni correction were performed to identify specific group differences. Data entry and statistical analysis were performed using Microsoft Excel^®^ (Microsoft Corporation, Redmond, WA, USA) and Python (version 3.9). The level of statistical significance was set at *p* < 0.05.

## 3. Results

### 3.1. Statistical Analysis of Primary Parameters

#### 3.1.1. Harvesting and Insertion Time

Harvesting time significantly varied between the different ARA techniques (Kruskal–Wallis test, *p* = 0.0025). The longest harvesting durations were recorded for the ICG group (mean ± SD: 51.6 ± 5.8 min; range: 41.3–60.1 min), followed by the IBB group (36.5 ± 10.8 min; range: 18.5–55.3 min). No harvesting was required for the ABB, CAD/CAM ABB, CAD/CAM TM, MS, and ST groups.

Insertion times also showed significant differences between the ARA techniques (Kruskal–Wallis test, *p* < 0.0001). Post-hoc pairwise comparisons revealed significant differences between ICGs and CAD/CAM ABBs, CAD/CAM TMs, and MSs and between IBBs and CAD/CAM ABBs, CAD/CAM TMs, and MSs (all *p* < 0.05 after Bonferroni correction). The longest insertion times were observed in the IBB group (50.1 ± 7.5 min; range: 34.8–60.5 min) and the ST group (47.3 ± 13.9 min; range: 27.9–69.3 min). The ICG group demonstrated a mean insertion time of 45.3 ± 8.6 min (range: 32.8–60.1 min). ABBs required 39.7 ± 5.9 min (range: 30.7–49.2 min), CAD/CAM ABBs 29.9 ± 5.6 min (range: 18.8–39.1 min), CAD/CAM TMs 28.5 ± 4.1 min (range: 22.3–36.4 min), and MSs 29.3 ± 6.9 min (range: 17.1–39.0 min) (Figure 8; Table 2).

#### 3.1.2. Vertical and Horizontal Bone Gain

Vertical bone gain significantly differed between the ARA techniques (Kruskal–Wallis test, *p* < 0.0001). Post-hoc pairwise comparisons revealed that ICGs and IBBs demonstrated significantly higher vertical bone gain compared to ABBs and CAD/CAM ABBs (all *p* < 0.05 after Bonferroni correction). ABBs and CAD/CAM ABBs also showed significantly lower vertical bone gain compared to CAD/CAM TMs, MSs, and the ST. The highest mean vertical bone gain was observed in the ICG group (5.6 ± 0.4 mm; range: 5.0–6.3 mm), followed by IBBs (4.4 ± 0.2 mm), CAD/CAM TMs (4.9 ± 1.0 mm), MSs (4.8 ± 0.6 mm), and the ST (4.7 ± 0.6 mm). CAD/CAM ABBs and ABBs showed comparable vertical bone gain, both around 3.0 ± 0.2 mm.

Horizontal bone gain also showed significant differences (Kruskal–Wallis test, *p* < 0.0001). Post-hoc pairwise comparisons demonstrated that ICGs and IBBs had significantly higher horizontal bone gain compared to ABBs and CAD/CAM ABBs (all *p* < 0.05 after Bonferroni correction). ABBs and CAD/CAM ABBs had significantly lower horizontal bone gain compared to CAD/CAM TMs, MSs, and STs. The ICG group showed the highest horizontal bone gain (5.6 ± 0.4 mm; range: 5.1–6.4 mm), followed by IBBs (5.2 ± 0.6 mm), MSs (5.2 ± 0.6 mm), and STs (4.9 ± 0.4 mm). CAD/CAM TMs reached a mean of 4.6 ± 0.5 mm, while CAD/CAM ABBs and ABBs showed 3.6 ± 0.3 mm and 3.5 ± 0.2 mm, respectively (Table 2).

#### 3.1.3. Graft Resorption

Resorption rates significantly differed between the ARA techniques (Kruskal–Wallis test, *p* < 0.0001). Post-hoc pairwise comparisons demonstrated significantly higher resorption in ICGs compared to all other techniques (all *p* < 0.05 after Bonferroni correction). CAD/CAM ABBs also showed significantly higher resorption compared to IBBs, ABBs, MSs, and the ST. MSs had the lowest resorption rates overall (5.1 ± 1.5%), while the highest resorption rates were observed in ICGs (25.9 ± 3.9%). IBBs and ABBs had similar, lower resorption rates around 9.9 ± 3.7% and 9.7 ± 2.3%, respectively. CAD/CAM TMs and the ST showed intermediate values (11.0 ± 3.0% and 7.2 ± 1.8%, respectively) (Figure 9; Table 2).

### 3.2. Descriptive Analysis of Secondary Parameters

#### 3.2.1. Donor Site Morbidity

Donor site morbidity was most frequently observed in the ICG group, where all patients (10/10, 100%) experienced postoperative complaints. The most commonly reported symptoms included pain and gait disturbances; no case of iliac crest fracture occurred. In contrast, the IBB group showed a lower incidence of donor site morbidity (5/10, 50%; here, the symptoms were pain, wound healing disturbances, and one case of temporary paesthesia). No donor site morbidity was observed in the other groups, ABB, CAD/CAM ABB, TM, MS, and ST.

#### 3.2.2. Dehiscence Rate

Dehiscence rates varied between the different ARA techniques. A rate of 10% (1/10 cases) was observed for ICGs, IBBs, ABBs, CAD/CAM ABBs, MSs, and the ST. The CAD/CAM TM group had the highest dehiscence rate, with 30% (3/10 cases). Even so, implant placement was reported to be possible in all cases.

#### 3.2.3. Removals

Removals (e.g., scaffold and screw removals) even differed between the different ARA techniques and grafts. Using ICGs, IBBs, ABBs, CAD/CAM ABBs, and the ST fixation screws were removed. CAD/CAM TMs required scaffold removal, and MSs did not need any removals, neither fixation screw nor scaffold removals.

#### 3.2.4. Biological and General Financial Costs

The highest biological costs were estimated for ICGs, followed by a moderate biological impact for IBBs and CAD/CAM TMs. In contrast, ABBs, CAD/CAM ABBs, MSs, and the allogeneic ST were associated with low biological costs.

General financial costs were highest for ICGs and CAD/CAM ABBs. Moderate costs were observed for IBBs, ABBs, CAD/CAM TMs, and STs, while MSs were associated with the lowest general financial cost among all ARA techniques.

## 4. Discussion

This study evaluated the clinical performance and limitations of different ARA techniques and grafting materials by analyzing cases of vertical augmentations exceeding 3 mm. There is no universally accepted definition of a “critical” vertical augmentation height across all ARA techniques and clinical contexts. However, in GBR, vertical augmentation beyond 3 mm is generally considered a limiting factor, and one-stage implant placement is typically not recommended in such cases [25]. The statistical analyses confirmed significant differences between the different augmentation techniques, particularly for harvesting and insertion times, vertical and horizontal bone gain, as well as graft resorption (all *p* < 0.05). These findings underscore the heterogeneity of different ARA techniques and highlight the need to carefully weigh procedural complexity and expected outcomes in clinical decision-making. The significant differences in vertical and horizontal bone gain, as well as in resorption and complication rates, further underscore that these outcomes should be carefully balanced with surgical time, biological cost, and patient morbidity in daily practice.

The ICG demonstrated the most significant vertical and horizontal bone gain (>5 mm in both dimensions), underlining its effectiveness in managing severely atrophic alveolar ridges. This finding aligns with previous studies, which reported vertical bone gains ranging from 8.5 to 9.4 ± 3.1 mm using ICGs [24,26]. However, the ICG was also associated with the highest resorption rates in this study (mean 25.9%; range: 19–32%), which is supported by previous findings, with resorption rates of 41.4 ± 19.6% after 12 months in cleft patients undergoing vertical ridge augmentation with ICG [27]. Analog to our findings, Mertens et al. reported a resorption rate of approximately 24% within six months following iliac crest grafting [26]. Other studies observed higher resorption when an ICG was used for onlay grafting compared to sinus floor elevation after 12 months of follow-up [28]. Sufficient graft incorporation appears to be completed in the atrophic maxilla after four months [29]. These findings suggest that graft type (e.g., onlay grafting vs. sinus floor elevation), anatomical location (maxilla vs. mandible), implant placement, and loading time influence ICG resorption.

Despite its regenerative potential, the ICG was associated with the highest donor site morbidity in this study (100%), such as postoperative pain, sensory disturbances, and gait impairment [30,31,32], contributing significantly to patient discomfort and overall patient cost. Complication rates at the recipient site have also been reported in over 30% of cases [26], which is notably higher than soft tissue complications such as dehiscence in GBR procedures (about 16.8%) [33]. In addition to high direct financial costs, such as hospitalization, the ICG may entail considerable indirect healthcare expenses, such as rehabilitation (e.g., physiotherapy) and loss of income due to work incapacity [34].

In contrast, the IBB demonstrated slightly lower vertical bone gain (mean 4.4 mm; range: 4.2–4.9) and reduced resorption rates (mean 9.9%), positioning the IBB as a viable alternative when autogenous grafts are clinically indicated. In contrast, other studies have reported significantly higher vertical and horizontal bone gain using autogenous shell techniques, with values of 7.6 ± 3.1 mm and 8.1 ± 1.6 mm, respectively. Vertical resorption was limited to 0.66 ± 0.38 mm after 12 months and was calculated to be approximately 11.4% after 10 years of follow-up, indicating favorable long-term volume stability [35]. Variations in bone gain and resorption may depend on whether a full-thickness IBB is used as onlay grafts or split for shell techniques. While IBB harvesting was associated with a markedly lower donor site morbidity (50%) compared to ICGs, it remains a time-consuming procedure. Additionally, it carries a risk of intraoperative nerve exposure, reported in approximately 4% of cases, which can result in transient or even permanent sensory disturbances [36,37,38,39]. In a comparative study, Heimes et al. reported that the IBB was associated with twice the rate of temporary nerve disturbances compared to an ABB in ridge augmentation. Additionally, postoperative swelling was found to be more pronounced in the IBB group [40]. Interestingly, Khoury et al. reported a mean harvesting time of 6.5 ± 2.5 min for IBBs using a standardized MicroSaw protocol, suggesting that specific instrumentation can significantly reduce surgical time and morbidity [36]. Contrasting results were reported for piezoelectric harvesting, with an average IBB harvesting time of 16.5 ± 2.7 min, substantially longer than the MicroSaw protocol. However, no differences in postoperative pain, swelling, or overall healing were found between the two techniques, suggesting comparable biological outcomes [41]. The significantly longer harvesting and insertion times associated with ICGs and IBBs observed in the present study emphasize their higher surgical complexity, which is consistent with the observed donor site morbidity. In contrast, CAD/CAM-based techniques and resorbable scaffolds demonstrated shorter surgical times and lower complication rates. These differences may be particularly relevant in patients with limited general health reserves or a high risk of donor site complications. Overall, donor site morbidity, harvesting time, and patient discomfort are strongly influenced by the surgeon’s experience and the selected surgical technique. In TMs, MSs, and the allogeneic ST, using autogenous bone chips for space filling may be associated with donor site morbidity and increased harvesting time, which was not considered in this study.

ABBs and CAD/CAM-manufactured ABBs demonstrated moderate vertical (about 3 mm) and horizontal (3–4 mm) bone gain in this study, with resorption rates ranging from 6–13% for ABBs and 12–25% for CAD/CAM ABBs after at least two years. A case report described a much lower resorption rate of 3–4.3% for CAD/CAM ABBs after six months, highlighting the potential for optimized outcomes in selected cases [42]. In addition to the limited number of cases, differences in resorption rates may also be attributed to the individual bone quality of the allogeneic donor material. Unlike allogeneic bone chips, allogeneic blocks are not pooled, which may contribute to variability in biological behavior. In a comparative study, Seidel et al. assessed ABBs and CAD/CAM ABBs regarding bone volume, gain, and stability. After six months, CAD/CAM ABBs showed slightly higher bone stability (87.6% ± 9.9%) compared to conventional ABBs (83% ± 14.5%), suggesting improved volumetric retention with patient-specific design [16]. This favorable outcome may be attributed to the precise fit of customized CAD/CAM ABBs, which potentially facilitates more efficient and higher-quality bone remodeling. Additionally, bone stability was generally higher in the maxilla compared to the mandible (91.6% vs. 79.5%), possibly due to differences in vascularity, bone density, or mechanical loading [16]. This may be explained by the fact that spongious CAD/CAM ABBs more closely resemble the trabecular structure of the maxilla, whereas a dense cortical bone layer characterizes the mandible. Compared to IBBs, CAD/CAM ABBs showed increased horizontal bone gain and reduced resorption when measured 1 mm below the alveolar crest after six months [43]. Despite their clinical advantages, resorption rates of allogeneic grafts remain a concern, underscoring the importance of careful patient selection. In the present study, insertion time for prefabricated ABBs was longer compared to customized CAD/CAM ABBs, consistent with previous reports indicating a mean time difference of approximately 52 min between the two techniques [16]. This observed difference may be attributed to the intraoperative complexity of defect-specific ABB customization by the surgeon using prefabricated ABBs, as well as to defect size and the number of CAD/CAM ABB elements inserted. A significant advantage of allogeneic bone grafts, particularly in extensive defects, is the complete elimination of donor site morbidity.

CAD/CAM TMs and MSs demonstrated a wide range of vertical and horizontal bone gain (mean 4.8–4.9 mm), suggesting that their rigid structure provides sufficient space maintenance for GBR and supports effective bone formation. This study’s mean vertical bone gain corresponds well with previously reported values (5.2 ± 1.6 mm; range 3.1–8.0 mm) [18]. Sagheb et al. reported a mean vertical and horizontal bone gain of 6.5 ± 1.7 mm and 5.5 ± 1.9 mm for CAD/CAM TMs, respectively, after 6 months [44]. Follow-up data demonstrated an implant survival rate of 97% in CAD/CAM TM-grafted areas and a sufficient marginal bone after 5 years, indicating its long-term stability [45]. As a limitation, CAD/CAM TM graft exposure was reported in about 33% of the cases during healing time without needing early graft removals [44]. Wound dehiscence may be attributed to the rigid TM structure. Soft tissue dehiscence remains a common challenge in ARA. Compared to poncho incisions, exposure rates were higher using crestal incisions (45.5% vs. 20%) [44]. Alternatively, deep mattress resorbable sutures can be used before crestal wound closure to relieve tension from the crestal wound closure and prevent dehiscence.

MSs had the lowest resorption rate (mean 5.1%), which may contribute to improved long-term stability of the grafted area. A possible reason could be MS biodegradability without graft removal, such as for CAD/CAM TMs, which is associated with a renewed detachment of the periosteum. MSs did present a low dehiscence rate, which may also be related to their biodegradability. The defect filling of CAD/CAM TMs and MSs is relevant, and there is no conclusion about the best filling possible, which is autogenous and allogeneic bone chips and particulate BS alone or in a mixture. However, using autogenous bone chips can be associated with donor-site morbidity.

The ST demonstrated a promising balance between bone gain (mean 4.7 mm vertically; mean 4.9 mm horizontally), insertion time (28–69 min), resorption (mean 7.2%), dehiscence rates (10%), and low biological costs, making it a suitable alternative to other ARA techniques [46,47,48]. Tunkel et al. compared the autogenous and allogeneic ST without differences between intra- and postoperative complications. A significantly reduced surgery time for allogeneic compared to autogenous shells, which is dependent on the surgeon’s experience, was found: 68.2 ± 24.8 min for allogeneic shells and 94.2 ± 38.5 up to 109 ± 32.8 min for autogenous shells, depending on the harvesting site [49]. Comparable to the current study, complication rates of about 8% for the allogeneic ST after a mean follow-up of 3.5 years were reported, mainly plate loss and dehiscence. Implant placement was possible in over 85% of complicated cases. The overall implant survival rate was >99% [50]. The biological and general financial costs are key factors for the ARA technique and graft selection. The ICG was associated with the highest biological and general financial costs due to its extensive surgical requirements and donor site morbidity. IBBs, CAD/CAM TMs, and the ST had medium biological costs, while ABBs, CAD/CAM ABBs, and MSs were the least biologically invasive. Even so, the evaluation of “biological cost” and “general financial cost” in our study was based on qualitative estimates and descriptive data from the participating centers. No standardized or validated economic or biological outcome measures were applied, which limits the comparability and generalizability of these results. Future studies should consider using validated health-economic or biological burden assessment tools to provide more robust and comparable data in this context.

A key strength of our study is the systematic integration of comprehensive statistical analyses, including post-hoc tests to delineate pairwise differences. However, the retrospective design and small group sizes limit the generalizability of these results, though this study also has several limitations. The retrospective study design limited the availability of detailed defect measurements (e.g., 3D volumetric defect analysis) and implant-specific data (dimensions and types), which were not consistently documented across centers. Another important consideration is the heterogeneity of individual patient factors (e.g., smoking status, anatomical site) that, while balanced across groups, may still influence outcomes and limit direct comparability. Furthermore, detailed data on the types of prosthetic restorations (e.g., single crowns, bridges, or full-arch rehabilitations) were not systematically documented and could therefore not be analyzed. The absence of a defined follow-up period limits the analysis of long-term outcomes, such as ARA success and implant survival. The clinical and radiological follow-up of the included cases extended to at least one year after implant placement, allowing for intermediate-term outcome analysis. However, five-year follow-up data were not available, which does not allow conclusions about long-term stability. Another limitation is the lack of systematic documentation of the soft tissue biotype in patients, which may influence technique selection, dehiscence rates, and clinical outcomes, but was not consistently available in the medical records of the participating centers. Further limiting factors are the low number of patients, different alveolar ridge defect sizes and locations, different surgeons, different particular grafts and bone substitutes used, and various soft-tissue management. Future prospective studies with larger cohorts and standardized data collection are essential to validate these findings and to further investigate long-term prosthetic outcomes and patient satisfaction.

## 5. Conclusions

The data demonstrate that the different ARA techniques and grafts significantly vary in terms of donor site morbidity, surgical time, bone gain, graft resorption, and associated costs. The ICG achieved the highest vertical and horizontal bone gain (~5.6 mm) but also had the longest harvesting time, the highest donor site morbidity (100%), and the highest resorption rates (~26%). In contrast, less invasive alternatives, such as CAD/CAM-guided TMs, MSs, ABBs, and the ST, demonstrated shorter surgical times, lower complication rates, and lower biological costs while still achieving substantial vertical bone regeneration (>3 mm in most cases). The significant differences between these ARA techniques underscore the importance of an individualized treatment plan. Selection of the most appropriate technique should consider defect size and location, anatomical limitations, surgical skill, and patient-specific factors, including risk of donor site complications and postoperative morbidity. While this study offers a comprehensive snapshot of current clinical practice across multiple centers, future prospective trials with standardized protocols, larger cohorts, and extended follow-up are needed to validate these findings and establish robust clinical guidelines.

## Figures and Tables

**Figure 1 jcm-14-04284-f001:**
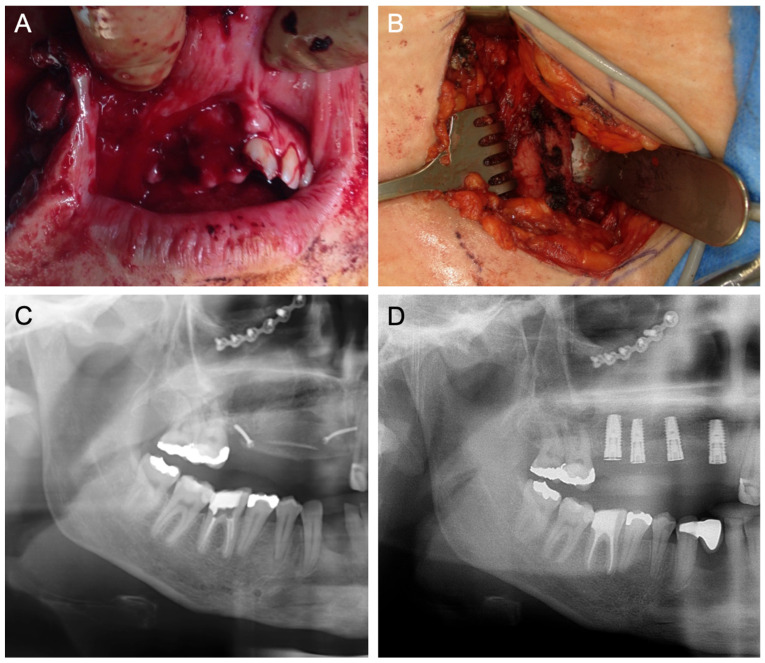
(**A**) Baseline following high-velocity trauma with midface fractures, alveolar ridge and tooth fractures of the right maxilla, loss of teeth and alveolar bone, and intra- and extraoral soft tissue wounds. (**B**) In an intraoperative situation, an ICG is harvested from the iliac crest for ARA of the right maxilla. (**C**) Panoramic X-ray demonstrates the right maxilla‘s inserted ICG. (**D**) Panoramic X-ray following the insertion of four implants in the right maxilla. Osteosynthesis plates are in situ from the surgical treatment of the midface fractures.

**Figure 2 jcm-14-04284-f002:**
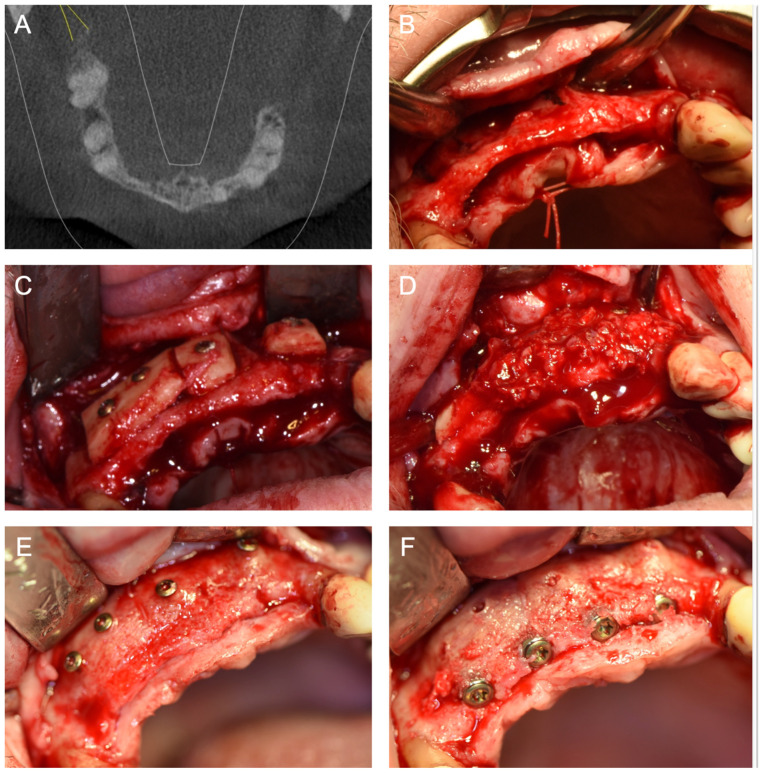
(**A**) Baseline CBCT scan and (**B**) intraoperative baseline demonstrating horizontal atrophy of the anterior maxilla after tooth loss. (**C**) Intraoperative situation after horizontal ARA with IBB harvested from the mandibular angle. (**D**) Placement of autogenous bone chips harvested from the mandibular angle and coverage of the augmented area with a porcine membrane (not shown). (**E**) Intraoperative situation six months after horizontal ARA with well-integrated IBB and a sufficient horizontal augmentation volume. Based on the position of the screw heads, a slight horizontal volume loss of the IBB can be inferred. (**F**) Full-guided insertion of four implants.

**Figure 3 jcm-14-04284-f003:**
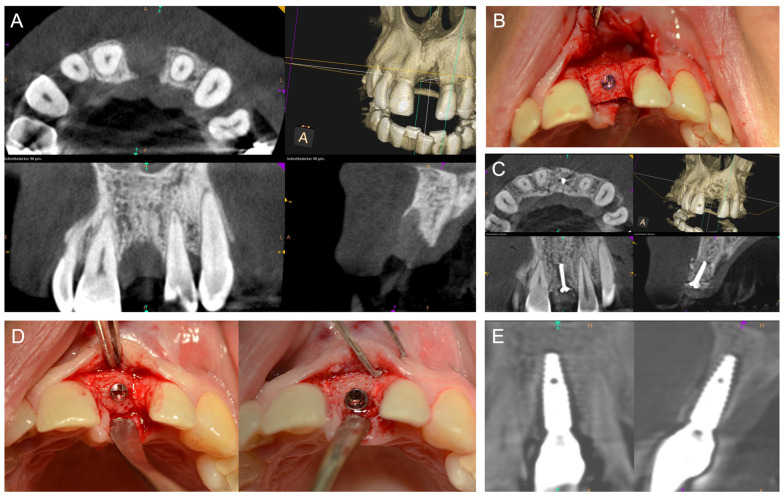
(**A**) Baseline CBCT scan demonstrating a vertical and horizontal alveolar ridge defect in the first incisor region on the left. (**B**) Intraoperative situation after placing an ABB, which was customized intraoperatively by the surgeon to the defect size. (**C**) A CBCT scan six months after surgery showed a well-integrated ABB. (**D**) Intraoperative situation six months postoperatively with implant placement. (**E**) Follow-up CBCT scan demonstrating the well-osseointegrated implant and an adequate peri-implant bone situation.

**Figure 4 jcm-14-04284-f004:**
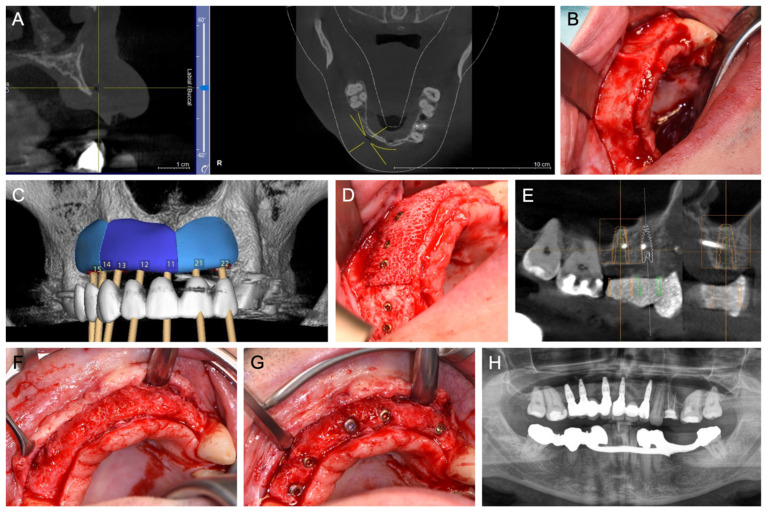
(**A**) Baseline CBCT scan and (**B**) intraoperative situation after tooth loss and horizontal alveolar ridge atrophy in the second premolar region on the right, to the second incisor region on the left side. (**C**) Digital planning of three CAD/CAM ABBs in this region for horizontal ridge augmentation. (**D**) An intraoperative situation involves the insertion of the three CAD/CAM ABBs for horizontal ridge augmentation and subsequent coverage with a porcine membrane (not shown). (**E**) CBCT scan six months after surgery shows well-integrated CAD/CAM ABB. (**F**) Intraoperative situation after removal of osteosynthesis screws and (**G**) fully-guided placement of five implants. (**H**) The panoramic X-ray shows adequate peri-implant bone conditions after prosthetic loading.

**Figure 5 jcm-14-04284-f005:**
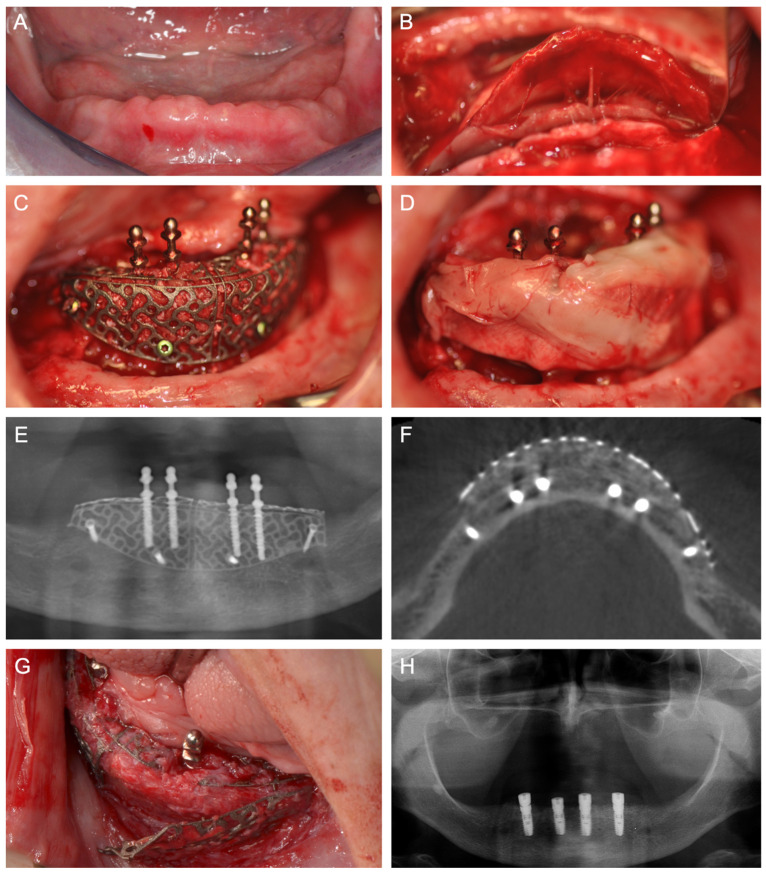
(**A**) Baseline of an edentulous lower jaw with a trembling comb. (**B**) Intraoperative situation of the edentulous lower jaw with an extensive bone deficit. (**C**) Placement of a CAD/CAM TM filled with a mixture of autogenous bone chips and a particular bovine BS. Additionally, four temporary implants were inserted for prosthetic interim restoration. (**D**) Coverage of the CAD/CAM TM with PRF membranes. (**E**) Postoperative panoramic X-ray demonstrating the filled CAD/CAM TM and the temporary implants. (**F**) CBCT scan six months after surgery with a sufficient augmentation volume in both dimensions. (**G**) An intraoperative situation occurred six months postoperatively during the removal of the CAD/CAM TM, and a well-integrated bone graft was demonstrated. (**H**) Panoramic X-ray after intraforaminal insertion of four implants.

**Figure 6 jcm-14-04284-f006:**
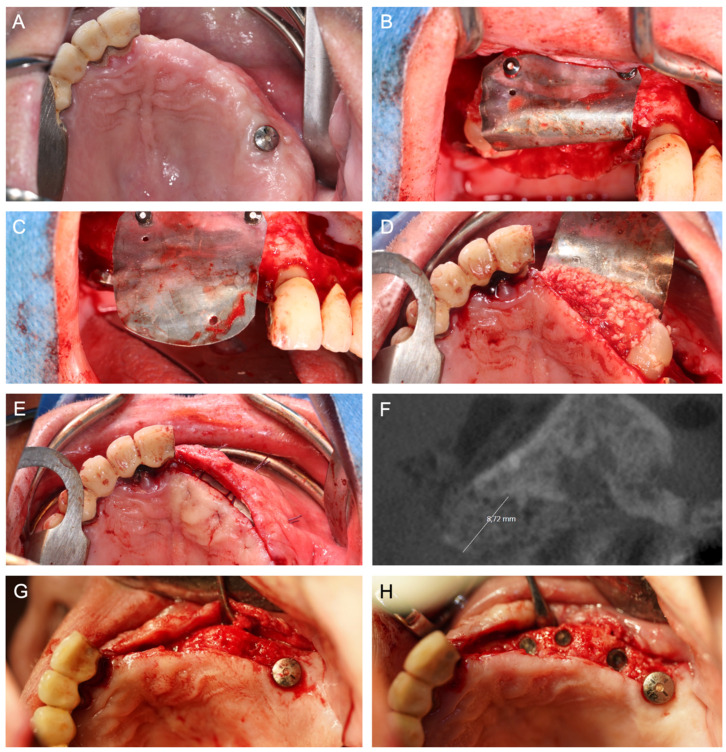
(**A**) Baseline with a toothless left upper jaw and vertical and horizontal alveolar ridge atrophy. (**B**,**C**) Intraoperative situation demonstrating placement and fixation of a biodegradable MS for ARA. (**D**) After adopting the MS, the defect is filled with a mixture of autogenous bone chips and allogeneic granules. (**E**) Deep mattress resorbable sutures are used before crestal wound closure to relieve tension from the crestal wound closure and prevent dehiscence. (**F**) A CBCT scan six months after surgery illustrates the augmented area with adequate vertical and horizontal bone gain. (**G**) Reentry six months postoperatively with a well-integrated bone graft and (**H**) insertion of three implants.

**Figure 7 jcm-14-04284-f007:**
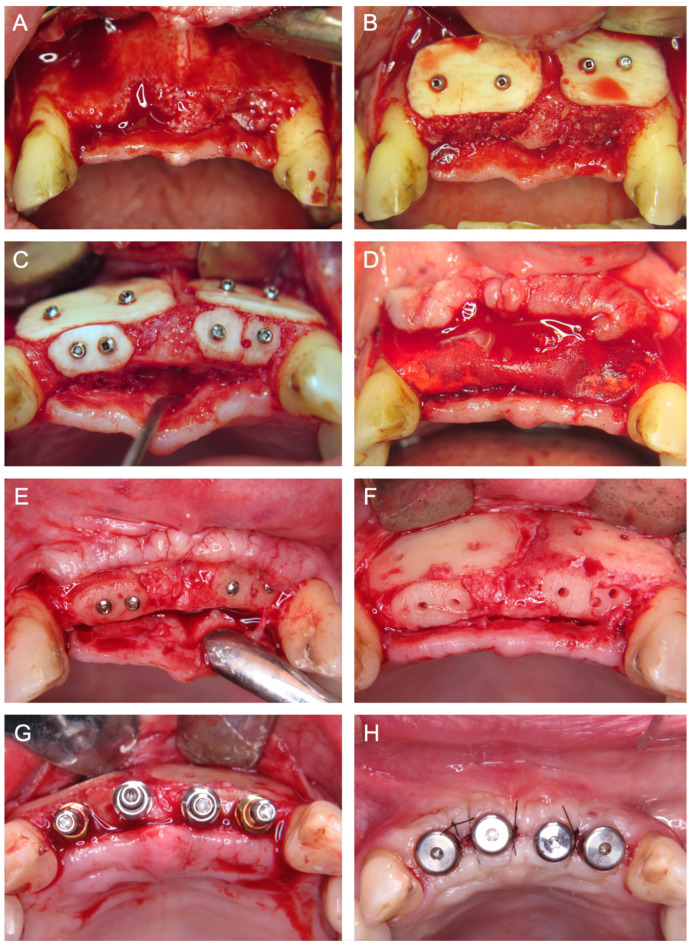
(**A**) Baseline after tooth loss and horizontal and vertical bone atrophy in the frontal second incisor region from left to right. (**B**) Insertion of two allogenic cortical bone plates for ST and defect filling with allogenic granules. (**C**) Crestal placement and screw fixation of two allogenic cortical bone plates. (**D**) Coverage of the augmented area with a porcine membrane. (**E**) Intraoperative situation six months after surgery and (**F**) after removal of the osteosynthesis screws demonstrating a well-integrated bone graft. (**G**) Insertion of four implants and (**H**) wound closure with healing abutments.

**Figure 8 jcm-14-04284-f008:**
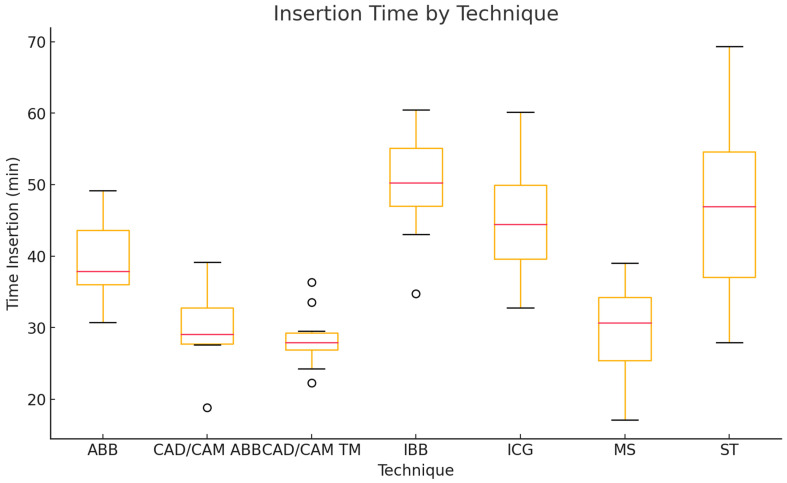
Boxplot of insertion time for each vertical alveolar ridge augmentation (ARA) technique.

**Figure 9 jcm-14-04284-f009:**
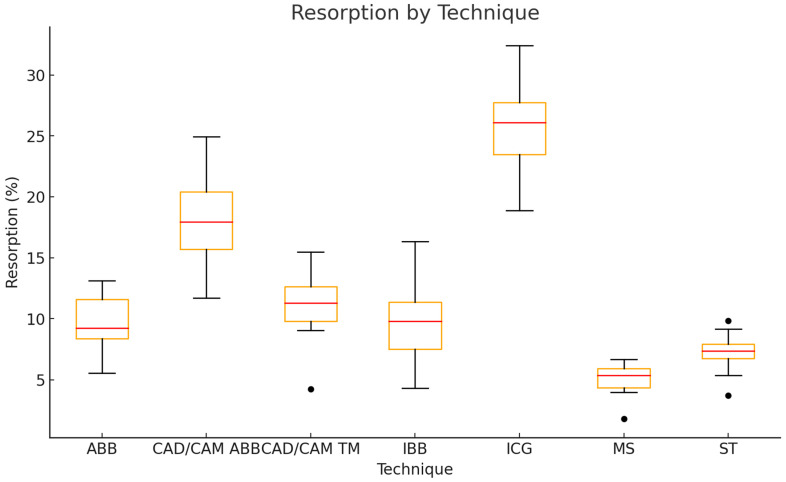
Boxplot of graft resorption rates for each ARA technique.

**Table 1 jcm-14-04284-t001:** Demographic and clinical characteristics of the patient cohorts for each vertical alveolar ridge augmentation (ARA) technique, including anatomical locations and implant placement approach.

Technique	Age (Years)	Sex Distribution	BMI	Smoking Status	Anatomical Location	Implant Placement
ICG	55.2 ± 4.7	5 Male/5 Female	25.4 ± 1.8	2 Smoker/8 Non-smoker	6 Maxilla/4 Mandible; 4 anterior/6 posterior	Staged placement; 2–4 implants/defect
IBB	54.4 ± 2.7	5 Male/5 Female	24.1 ± 1.6	1 Smoker/9 Non-smoker	5 Maxilla/5 Mandible; 5 anterior/5 posterior	Staged placement; 1–4 implants/defect
ABB	56.1 ± 3.6	5 Male/5 Female	25.9 ± 1.4	2 Smoker/8 Non-smoker	5 Maxilla/5 Mandible; 5 anterior/5 posterior	Staged placement; 1–3 implants/defect
CAD/CAM ABB	56.2 ± 2.6	5 Male/5 Female	25.4 ± 1.6	2 Smoker/8 Non-smoker	6 Maxilla/4 Mandible; 4 anterior/6 posterior	Staged placement; 1–5 implants/defect
CAD/CAM TM	52.0 ± 3.2	5 Male/5 Female	25.4 ± 1.1	2 Smoker/8 Non-smoker	4 Maxilla/6 Mandible; 4 anterior/6 posterior	Staged placement; 1–3 implants/defect
MS	55.5 ± 4.9	5 Male/5 Female	24.8 ± 1.6	1 Smoker/9 Non-smoker	6 Maxilla/4 Mandible; 6 anterior/4 posterior	Staged placement; 1–3 implants/defect
ST	54.7 ± 4.3	5 Male/5 Female	25.1 ± 0.9	3 Smoker/7 Non-smoker	5 Maxilla/5 Mandible; 4 anterior/6 posterior	Staged placement; 1–4 implants/defect

**Table 2 jcm-14-04284-t002:** Quantitative parameters (mean ± standard deviation) and ranges for each vertical alveolar ridge augmentation (ARA) technique.

Technique	Time Harvesting (min)	Time Insertion (min)	Vertical Gain (mm)	Horizontal Gain (mm)	Resorption (%)
ICG	51.6 ± 5.8 (range: 41.3–60.1)	45.3 ± 8.6 (range: 32.8–60.1)	5.6 ± 0.4 (range: 5.0–6.3)	5.6 ± 0.4 (range: 5.1–6.4)	25.9 ± 3.9 (range: 18.9–32.4)
IBB	36.5 ± 10.8 (range: 18.5–55.3)	50.1 ± 7.5 (range: 34.8–60.5)	4.4 ± 0.2 (range: 4.2–4.9)	5.2 ± 0.6 (range: 3.7–6.0)	9.9 ± 3.7 (range: 4.3–16.3)
ABB	None	39.7 ± 5.9 (range: 30.7–49.2)	3.0 ± 0.1 (range: 2.7–3.3)	3.5 ± 0.2 (range: 3.2–3.9)	9.7 ± 2.3 (range: 5.6–13.1)
CAD/CAM ABB	None	29.9 ± 5.6 (range: 18.8–39.1)	3.0 ± 0.2 (range: 2.7–3.3)	3.6 ± 0.3 (range: 3.2–4.2)	18.2 ± 3.9 (range: 11.7–24.9)
CAD/CAM TM	None	28.5 ± 4.1 (range: 22.3–36.4)	4.9 ± 1.0 (range: 3.2–6.2)	4.6 ± 0.5 (range: 4.0–5.4)	11.0 ± 3.0 (range: 4.3–15.5)
MS	None	29.3 ± 6.9 (range: 17.1–39.0)	4.8 ± 0.6 (range: 3.8–6.0)	5.2 ± 0.6 (range: 4.1–6.1)	5.1 ± 1.5 (range: 1.8–6.7)
ST	None	47.3 ± 13.9 (range: 27.9–69.3)	4.7 ± 0.6 (range: 3.9–5.8)	4.9 ± 0.4 (range: 4.2–5.3)	7.2 ± 1.8 (range: 3.7–9.8)

## Data Availability

Data will be available upon reasonable request.

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
