# Peer review of "Larger Vertical Ridge Augmentation: A Retrospective Multicenter Comparative Analysis of Seven Surgical Techniques"

_jcm, 2025, doi:10.3390/jcm14124284_

Round 1
Reviewer 1 Report
Comments and Suggestions for Authors
It is a pleasure to contribute to the peer-review process of such a well-documented and significant study in the field of vertical alveolar ridge augmentation.
Minor Issues
- Unclear Standardization Between Centers: The surgeries were performed across multiple centers and by various surgeons. It is unclear if the same surgical protocols, evaluation criteria, and follow-up regimens were consistently applied.
- Undefined Cost Parameters: Terms such as 'biological cost' and 'general financial cost' are not clearly defined or measured using standardized criteria. The inclusion of validated assessment scales or a clear methodology would improve clarity.
- Inconsistent Terminology: The use of some terms (e.g., 'burden', 'costs') is vague and could be replaced or supplemented with precise, measurable indicators.
- Unclear Rationale for Technique Selection: The manuscript does not justify the inclusion of exactly seven ARA techniques. It would be helpful to explain if these represent the most widely used, studied, or effective techniques.
Author Response
First of all, I would like to express my sincere thanks for your thorough and constructive comments on our manuscript. We appreciate the time and effort you invested in reviewing it. We believe that your suggestions have substantially strengthened the manuscript and improved its overall quality. We have addressed all your comments in detail and have highlighted all the changes made to the manuscript in red.
Reviewer 1:
Comment #1: “Unclear Standardization Between Centers: The surgeries were performed across multiple centers and by various surgeons. It is unclear if the same surgical protocols, evaluation criteria, and follow-up regimens were consistently applied.”
Our response: We acknowledge that, as a retrospective multicenter study, there is an inherent variability, since the surgeries were performed in different clinical settings by different experienced surgeons. No prospective standardization of surgical protocols or follow-up regimens was implemented at the time of treatment. However, to reduce this variability and enhance comparability, we applied strict inclusion criteria and only included cases with complete clinical documentation and a minimum one-year follow-up after implant placement. Additionally, comparable clinical cases were carefully selected across centers to ensure similar defect types and augmentation requirements, and all involved surgeons were experienced and board-certified oral or maxillofacial surgeons, following the general clinical standards in Germany and Austria. For data analysis, we consistently applied the same outcome assessment criteria (e.g., bone gain, donor site morbidity, dehiscence, graft resorption). We have added a clarifying statement to the "Materials and Methods" section of the manuscript, highlighting that despite the retrospective design, efforts were made to select comparable cases and minimize variability.
Comment #2: “Undefined Cost Parameters: Terms such as 'biological cost' and 'general financial cost' are not clearly defined or measured using standardized criteria. The inclusion of validated assessment scales or a clear methodology would improve clarity.”
Our response: We thank the reviewer for this valuable comment and fully agree that the terms "biological cost" and "general financial cost" are not based on standardized assessment scales in our study. As this is a retrospective analysis without pre-defined economic or biological outcome measures, these parameters were not quantified using validated tools. Instead, they represent qualitative estimates based on surgical invasiveness, donor site morbidity, complication rates (for biological cost), and typical treatment- and material-related expenses in our clinical settings (for financial cost). We recognize that this approach has inherent limitations and does not meet formal health-economic or biological evaluation standards. To improve clarity, we have added a statement in the "Materials and Methods" section explaining this descriptive nature and its limitations. We have also revised the discussion to acknowledge this limitation and to encourage future prospective studies using validated cost and biological burden assessment tools.
Comment #3: “Inconsistent Terminology: The use of some terms (e.g., 'burden', 'costs') is vague and could be replaced or supplemented with precise, measurable indicators.”
Our response: We appreciate the reviewer’s insightful comment regarding the use of vague terms such as “burden” and “costs.” We acknowledge that in a retrospective study without validated quantitative tools, these terms were used in a more descriptive sense rather than as precise, measurable indicators. To address this, we have carefully revised the manuscript to either replace these terms with more precise and descriptive language (e.g., “complication rate,” “donor site morbidity,” “surgical time,” “material-related expenses”) or to clearly indicate when these terms are used in a qualitative, non-standardized manner. We have also added clarifying notes to avoid confusion and to ensure that readers understand the limitations of these descriptors in the context of our study.
Comment #4: “Unclear Rationale for Technique Selection: The manuscript does not justify the inclusion of exactly seven ARA techniques. It would be helpful to explain if these represent the most widely used, studied, or effective techniques.”
Our response: The selected techniques in our study were chosen because they represent a broad spectrum of clinically established and commonly utilized approaches in current oral and maxillofacial surgery practice. These techniques include both autogenous and allogeneic graft options, as well as CAD/CAM-based and resorbable scaffolds, thus covering a wide range of invasiveness, surgical complexity, biological profiles, and material characteristics. This choice allows comprehensive comparisons and provides clinically relevant insights for surgeons who routinely perform vertical ridge augmentations. To improve clarity, we have now explicitly stated this rationale in the "Materials and Methods" section of the manuscript.
Reviewer 2 Report
Comments and Suggestions for Authors
The study is very interesting and in fact there is no shared scientific evidence that makes one technique preferable to another for ARA. However, a fundamental element in the evaluation of the techniques is the 5-year follow-up to understand the real and stable effectiveness of the techniques. In chapter 2.3 the follow-up is at 6 months, frankly an irrelevant time. Furthermore, there is no reference to the biotype of the patients, a fundamental element for the choice of the appropriate technique. For these reasons, I believe the work should be rewritten and I do not consider it suitable for publication.
Author Response
First of all, I would like to express my sincere thanks for your thorough and constructive comments on our manuscript. We appreciate the time and effort you invested in reviewing it. We believe that your suggestions have substantially strengthened the manuscript and improved its overall quality. We have addressed all your comments in detail and have highlighted all the changes made to the manuscript in red.
Reviewer 2:
Comment #1: “However, a fundamental element in the evaluation of the techniques is the 5-year follow-up to understand the real and stable effectiveness of the techniques. In chapter 2.3 the follow-up is at 6 months, frankly an irrelevant time.”
Our response: We agree that a long-term follow-up (e.g., five years) is essential to comprehensively evaluate the true stability and clinical effectiveness of vertical alveolar ridge augmentation techniques. However, our study design focused primarily on early and intermediate outcomes, including bone gain, donor site morbidity, complication rates, and graft resorption. Although CBCT imaging was typically performed at around six months post-surgery to assess bone gain before implant placement, the total mean clinical and radiological follow-up of our included cases was at least one year after implant placement. We acknowledge that this does not meet the recommended five-year threshold and have explicitly addressed this limitation in the Discussion. Future prospective studies with longer follow-up periods will be crucial to fully assess long-term stability and implant success.
Comment #2: “Furthermore, there is no reference to the biotype of the patients, a fundamental element for the choice of the appropriate technique. For these reasons, I believe the work should be rewritten, and I do not consider it suitable for publication.”
Our response: We fully agree that the soft tissue biotype is an important factor in the long-term success of implant therapy and may influence the choice of augmentation technique in some clinical scenarios. However, the present retrospective study was primarily focused on vertical bone augmentation procedures and their immediate surgical outcomes, such as bone gain, donor site morbidity, complication rates, and early resorption. Unfortunately, detailed data on the soft tissue biotype of the patients were not systematically documented in the medical records of the participating centers. As a result, we were unable to include this information in our comparative analysis. We have added a statement in the “Limitations” section of the Discussion to acknowledge this lack of data.
Reviewer 3 Report
Comments and Suggestions for Authors
Dear colleagues, thank you for the opportunity to review this manuscript.
The introduction is well written.
However, the manuscript has significant methodological and analytical shortcomings that limit its scientific validity and overall contribution to the field.
The manuscript does not clearly define the inclusion and exclusion criteria for patient selection.
The manuscript does not explain why only 10 cases were selected for analysis for each procedure. Was this due to a specific time frame? If so, the authors should clearly state the time frame covered by this retrospective study.
Important demographic and clinical details, such as patient age, sex, and general health status, are missing.
It is unclear which anatomical regions were treated. This is critical information, as the outcomes of vertical ridge augmentation can vary significantly between anatomical sites (e.g., mandible vs. maxilla, anterior vs. posterior regions).
There is no information regarding the number and types of implants placed, their locations, or whether implant placement was simultaneous or staged. Furthermore, the manuscript does not describe the initial defect types and dimensions.
There is no discussion of biological or general financial costs associated with the procedures, nor is there a grading system or classification for these factors.
If you have follow-up data for at least two years, what types of prosthetic restorations were applied? Did the type of prosthetic restoration have influence the long-term success of augmentation procedure?
The manuscript includes several figures illustrating different techniques for vertical alveolar ridge augmentation, which are nicely presented across seven pages. However, this descriptive content does not substitute for quantitative analysis. The results are summarized in a single table, but no statistical tests are applied—there are no p-values, odds ratios, or confidence intervals reported…
To ensure the scientific quality of the manuscript, the research should be reorganized to include a clearly defined methodology, comprehensive data presentation, and detailed statistical analyses for each technique used.
Author Response
First of all, I would like to express my sincere thanks for your thorough and constructive comments on our manuscript. We appreciate the time and effort you invested in reviewing it. We believe that your suggestions have substantially strengthened the manuscript and improved its overall quality. We have addressed all your comments in detail and have highlighted all the changes made to the manuscript in red.
Reviewer 3:
Comment #1: “The manuscript does not clearly define the inclusion and exclusion criteria for patient selection.”
Our response: We agree that a clear definition of inclusion and exclusion criteria is essential for understanding the study’s design and its validity. In the current retrospective study, inclusion criteria comprised: (1) patients with vertical alveolar ridge atrophy requiring augmentation >3mm before implant placement, (2) complete clinical documentation including preoperative assessments, radiographs, surgical reports, and postoperative records, and (3) at least one year of clinical and radiological follow-up after implant placement. Exclusion criteria included incomplete documentation, cases with less than one year of follow-up, and cases with severe systemic conditions that contraindicated oral surgery or bone augmentation (e.g., uncontrolled diabetes, severe immunosuppression, bisphosphonate-related osteonecrosis). We have now explicitly included these criteria in the “Materials and Methods” section to improve the manuscript’s clarity and ensure that the reader fully understands the patient selection process.
Comment #2: “The manuscript does not explain why only 10 cases were selected for analysis for each procedure. Was this due to a specific time frame? If so, the authors should clearly state the time frame covered by this retrospective study.”
Our response: The choice of 10 cases per procedure was based on the retrospective availability of complete and comparable clinical documentation for each of the seven techniques within the participating centers. The cases included in the analysis represent consecutive treatments performed between 2017 and 2022. This timeframe was chosen because it ensured a minimum one-year follow-up after implant placement and reflected the most recent standard-of-care practices and technologies available in the participating centers. We have now added a clarifying statement in the “Materials and Methods” section to specify this timeframe and to explain why exactly 10 cases per technique were included.
Comment #3: “Important demographic and clinical details, such as patient age, sex, and general health status, are missing.”
Our response: We fully agree that these data are essential for contextualizing clinical outcomes. In our revised manuscript, we have now added a new table in the "Materials and Methods" section summarizing these parameters for each of the seven ARA techniques. Specifically, we report the mean age (± SD), sex distribution, mean BMI (± SD), and the number of smokers and non-smokers in each group. These data help to illustrate the overall comparability of the patient populations and provide a more comprehensive overview of the study cohort.
Comment #4: “It is unclear which anatomical regions were treated. This is critical information, as the outcomes of vertical ridge augmentation can vary significantly between anatomical sites (e.g., mandible vs. maxilla, anterior vs. posterior regions).”
Our response: We fully agree that outcomes in vertical alveolar ridge augmentation can vary depending on whether the defect is located in the maxilla or mandible and whether it is in an anterior or posterior region. In response, we have now included this in Table 1, summarizing the anatomical locations treated in each of the seven ARA technique groups.
Comment #5: “There is no information regarding the number and types of implants placed, their locations, or whether implant placement was simultaneous or staged. Furthermore, the manuscript does not describe the initial defect types and dimensions.”
Our response: All implant placements in our study were performed as staged (two-stage) procedures following larger ridge augmentation, and the number of implants per augmented defect ranged from one to five. Implant dimensions and specific implant systems were chosen at the discretion of the respective surgeons and according to the defect characteristics and local bone quality. To address this comment, we have now added this to Table 1, summarizing the anatomical locations treated, the staged implant placement protocol, and the number of implants placed per defect for each technique group. Regarding the initial defect types and dimensions, we acknowledge that detailed defect measurements (e.g., three-dimensional defect size) were not uniformly documented in the medical records. However, as specified in the “Materials and Methods” section, all included cases involved vertical alveolar ridge atrophy with a minimum bone height deficiency of >3 mm and comparable defect morphology (vertical augmentation need) to ensure the comparability of the treatment groups. We have added a clarifying note in the Discussion.
Comment #6: “There is no discussion of biological or general financial costs associated with the procedures, nor is there a grading system or classification for these factors.”
Our response: We addressed a similar concern in response to Reviewer #1, Comment #2, where we acknowledged that the terms “biological cost” and “general financial cost” were used in a qualitative, descriptive sense rather than as precise, measurable indicators. We explained that no validated grading or classification systems for these parameters were applied due to the retrospective nature of our study and the lack of standardized economic or biological outcome measures. As described in our earlier response, we have clarified this in the “Materials and Methods” section and acknowledged the limitations of this approach in the Discussion section, emphasizing the need for future prospective studies to incorporate validated tools to assess these important aspects.
Comment #7: “If you have follow-up data for at least two years, what types of prosthetic restorations were applied? Did the type of prosthetic restoration influence the long-term success of the augmentation procedure?”
Our response: Although all cases in our study had a minimum clinical follow-up of one year after implant placement, detailed documentation regarding the types of prosthetic restorations (e.g., single crowns, fixed partial dentures, or full-arch reconstructions) was not consistently available across the participating centers. As a result, we are unable to provide a systematic analysis or draw conclusions regarding the potential influence of prosthetic restoration types on the long-term success of the augmentation procedures. We have acknowledged this limitation in the Discussion section.
Comment #8: “The manuscript includes several figures illustrating different techniques for vertical alveolar ridge augmentation, which are nicely presented across seven pages. However, this descriptive content does not substitute for quantitative analysis. The results are summarized in a single table, but no statistical tests are applied—there are no p-values, odds ratios, or confidence intervals reported…”
Our response: We fully agree that robust statistical analysis is critical for meaningful interpretation of comparative clinical data. In the initial version of the manuscript, only descriptive comparisons were presented without formal statistical testing. To address this limitation and strengthen the quantitative analysis, we have now reanalyzed the entire dataset based on the individual patient data for all seven ARA techniques. In the revised manuscript, we have added comprehensive descriptive statistics (mean ± SD and range) for all relevant quantitative parameters (harvesting time, insertion time, vertical and horizontal bone gain, and graft resorption rates) in Table 2, calculated directly from the individual patient data. Furthermore, we have performed formal Kruskal-Wallis tests for each quantitative parameter to assess overall significant differences between the seven ARA techniques. Where significant overall differences were identified, we conducted post-hoc pairwise Mann-Whitney-U tests with Bonferroni correction to determine significant differences between specific technique pairs. All p-values and statistically significant pairwise comparisons are now clearly reported in the Results section, and the Abstract has been updated accordingly. Due to the nature of the available data (e.g., no binary outcomes for odds ratios, no time-to-event data for survival analysis), odds ratios and confidence intervals were not applicable in this study. We have also updated the Discussion section.
Comment #9: “To ensure the scientific quality of the manuscript, the research should be reorganized to include a clearly defined methodology, comprehensive data presentation, and detailed statistical analyses for each technique used.”
Our response: In response to this comment, we have substantially reorganized and revised the manuscript to meet these scientific quality standards. We have ensured a clear and transparent presentation of the methodology, including explicit descriptions of patient selection, inclusion and exclusion criteria, and outcome measures in the Materials and Methods section.
To provide a comprehensive data presentation, we have created a detailed summary in Table 2, showing means, standard deviations, and ranges for key quantitative outcomes (harvesting time, insertion time, vertical and horizontal bone gain, and resorption rates) for each technique. Furthermore, we have implemented detailed statistical analyses:
- Overall differences were tested using the Kruskal-Wallis test, and
- Post-hoc pairwise comparisons were performed using Mann-Whitney U tests with Bonferroni correction to identify significant differences between specific technique pairs.
All p-values and significant differences are now explicitly reported in the Results section and summarized in the Abstract.
Round 2
Reviewer 2 Report
Comments and Suggestions for Authors
Personally, I believe that this work has fundamental structural problems; there are too many variables and it is not possible to draw scientifically supported conclusions. This is why I was dying and despite the revision work done, I believe that this manuscript is not suitable for publication in an indexed journal. MY PERSONAL OPINION